# Prediction Model for Pre-Eclampsia Using Gestational-Age-Specific Serum Creatinine Distribution

**DOI:** 10.3390/biology12060816

**Published:** 2023-06-04

**Authors:** Jieun Kang, Sangwon Hwang, Taesic Lee, Kwangjin Ahn, Dong Min Seo, Seong Jin Choi, Young Uh

**Affiliations:** 1Department of Obstetrics and Gynecology, Yonsei University Wonju College of Medicine, Wonju 26426, Republic of Korea; k131730@hanmail.net (J.K.); choisj@yonsei.ac.kr (S.J.C.); 2Department of Precision Medicine, Yonsei University Wonju College of Medicine, Wonju 26426, Republic of Korea; arsenal@yonsei.ac.kr; 3Division of Data-Mining and Computational Biology, Institute of Global Health Care and Development, Wonju 26426, Republic of Korea; ddasic123@yonsei.ac.kr; 4Department of Family Medicine, Yonsei University Wonju College of Medicine, Wonju 26426, Republic of Korea; 5Department of Laboratory Medicine, Yonsei University Wonju College of Medicine, Wonju 26426, Republic of Korea; kjahn123@yonsei.ac.kr; 6Department of Medical Information, Yonsei University Wonju College of Medicine, Wonju 26426, Republic of Korea; dmseo@yonsei.ac.kr

**Keywords:** pre-eclampsia, pregnancy, creatinine, gestational age, renal hyperfiltration

## Abstract

**Simple Summary:**

The prediction of pre-eclampsia (PE) is a crucial task both medically and socioeconomically. Recently, several biomarkers have been developed with clinically promising results. However, the currently identified markers face the challenges of their applicability in the clinical settings due to factors such as cost and measurement platform. According to our study, incorporating serum creatinine (SCr) levels that can be easily derived from a real-world hospital database along with prior knowledge of hyperfiltration, which is a kidney-specific physiological adaptation in pregnancy, improved PE prediction significantly. The model developed in this study is practical and can be easily applied in primary care settings without requiring significant hospital database upgrades.

**Abstract:**

Pre-eclampsia (PE) is a pregnancy-related disease, causing significant threats to both mothers and babies. Numerous studies have identified the association between PE and renal dysfunction. However, in clinical practice, kidney problems in pregnant women are often overlooked due to physiologic adaptations during pregnancy, including renal hyperfiltration. Recent studies have reported serum creatinine (SCr) level distribution based on gestational age (GA) and demonstrated that deviations from the expected patterns can predict adverse pregnancy outcomes, including PE. This study aimed to establish a PE prediction model using expert knowledge and by considering renal physiologic adaptation during pregnancy. This retrospective study included pregnant women who delivered at the Wonju Severance Christian Hospital. Input variables, such as age, gestational weeks, chronic diseases, and SCr levels, were used to establish the PE prediction model. By integrating SCr, GA, GA-specific SCr distribution, and quartile groups of GA-specific SCr (GAQ) were made. To provide generalized performance, a random sampling method was used. As a result, GAQ improved the predictive performance for any cases of PE and triple cases, including PE, preterm birth, and fetal growth restriction. We propose a prediction model for PE consolidating readily available clinical blood test information and pregnancy-related renal physiologic adaptations.

## 1. Introduction

Pre-eclampsia (PE) is defined as new-onset hypertension (HTN) and multiorgan damage, including proteinuria, diagnosed after 20 weeks of gestation [1]. PE is a significant pregnancy-related complication that contributes considerably to maternal and neonatal morbidity and mortality. It is the second most important cause of maternal death [2]. Each year, it causes over 70,000 maternal deaths, over 500,000 preterm births (PTB) and is related to 2% to 8% of all pregnancies worldwide [3,4]. PE has non-negligible incidence and causes further systemic complications and lifelong sequelae for mothers and their fetuses [5]. Severe PE is a leading cause of maternal morbidity (e.g., stroke and renal failure) and adverse pregnancy outcomes, such as PTB, fetal growth restriction (FGR), and intrauterine fetal death [6].

Prediction or early detection of PE is crucial for ensuring the safety of both pregnant women and their babies. In actual clinical practice, pharmacological interventions (e.g., low-dose aspirin at ≤16 weeks of gestation) to pregnant women categorized as high-risk for PE have been shown to prevent progression to severe and preterm PE [7]. The development of a prediction model to screen for high-risk PE pregnant women has significant medical utility in facilitating targeted surveillance and timely delivery [8]. Recent studies have identified biomarkers to predict or detect PE status and develop prediction model using combinations of clinical and laboratory variables [8,9]. However, the application of the aforementioned achievements (biomarkers and prediction models) is challenging in the primary clinical setting, due to the scarcity of diagnostic equipment and cost-related issues.

PE presents with HTN and multiple variable forms of organ failures, including dysfunctions of the kidney, brain, liver, and lungs, and abnormalities of hematologic clotting system [6,10]. Moreover, proteinuria, the cornerstone of the disease, results from increased renal tubular permeability [4]. PE is the most common glomerular-based kidney disease worldwide [11]. There is a widely established consensus regarding the association between kidney disease and PE [11,12]. However, there is a lack of detailed research analyzing the relationship between serum creatinine (SCr) levels and the prevalence and incidence of PE [13,14]. Furthermore, an Iranian study analyzed serum markers in approximately 450 patients with PE and demonstrated that SCr was a predictive marker of PE severity [15].

The renal physiologic alteration is critical for a favorable pregnancy outcome. PE shows variable degrees of renal dysfunction, and normal physiologic accommodation is not being conducted adequately. Harel et al. [16] collected approximately 362,000 measurements of SCr from 243,534 women and established the gestational age (GA)-specific SCr distribution embracing the renal physiologic adaptation in normal pregnancy. Moreover, they suggested that the dysregulation of pregnancy-related hyperfiltration, referred to as blunted glomerular hyperfiltration, was related to subsequent adverse maternal outcomes, including severe maternal morbidity [17]. Multidisciplinary expert groups at the Seoul National University suggested that a decrease in kidney function based on the midterm estimated glomerular filtration rate (GFR) was related to high-risk adverse pregnancy outcomes [14]. Based on these previous studies, our team found that the GA-specific SCr distribution can reflect adverse pregnancy outcomes [18]. The current study aimed to establish the PE prediction model using kidney function. Moreover, we evaluated whether the GA-specific SCr distribution proposed by Harel et al. [16] improves the predictive performance of PE status.

## 2. Materials and Methods

### 2.1. Study Participants

The initial dataset was constructed using patients’ medical records and included women who delivered at the Department of Obstetrics, Wonju Severance Christian Hospital (WSCH) since 2002 [18]. This automatic platform was implemented to supplement the initially constructed dataset with medical and laboratory data [18,19]. In detail, candidate pregnancy cases were curated as the initial dataset (i.e., the patients’ case note). Then, the automatic platform was processed to add the SCr levels and their measurement times to the initial dataset by a database administrator team (Appendix A). Next, a domain expert team, including an obstetrician and a family medicine physician, manually added the chronic disease status (e.g., HTN and diabetes (DM)) and adverse pregnancy outcomes (Appendix A). Blood tests such as SCr were conducted during early, mid-, and late periods of pregnancy. The timing of initial visits for pregnant women varies individually. In cases that are referred from a primary local clinic, the first blood test is conducted during the second or third trimester. Most pregnant women undergo a systematic blood test during the third trimester of pregnancy before childbirth. Data regarding the gestational weeks (GWs) of expectant mothers were recorded in both electronic medical records (EMRs) and manually curated pregnancy case notes. Moreover, during every single medical visit, information about each patient’s pregnancy was recorded individually in the EMR. Patients’ case notes, manually collected by obstetricians, contained information on the delivery dates of mothers and their corresponding GWs. By cross-referencing the delivery data with the timing of SCr measurements, it was possible to accurately estimate the GW at the time of the blood test. Finally, data scientists preprocessed the raw dataset (i.e., the initial dataset constructed manually and the automatic platform) into a normalized dataset to construct the prediction model. A previous study implemented a sample-based measurement of SCr and found that most pregnant women performed one or two laboratory examinations [18]. Moreover, in this study, we included all SCr examination results. In other words, regardless of the SCr levels obtained from the same patient, if the test timing was different, we considered them as different measurements.

The participants were pregnant women who delivered singleton pregnancies at more than 20 weeks of gestation. Afterward, we excluded women who were aged less than 16 years or more than 50 years at delivery or had incomplete data on SCr levels and their measurement times. Finally, 10,126 measurements of SCr levels were used to construct the adverse pregnancy outcome prediction models.

This was an observational, retrospective study without an intervention or experiment; therefore, the requirement of informed consent was waived. This study protocol and the waiver of informed consent were approved by the Institutional Review Board of the WSCH (CR321084). All analyses in this study were conducted in accordance with the principles of the Declaration of Helsinki.

### 2.2. Selection of Predictors Related to PE Status

The feature selection is a crucial step for the establishment of the prediction model. Several studies initially selected disease or phenotype-related features or predictors by expert knowledge and literature-based search [20,21,22]. Based on the expert knowledge and our previous study [18], we pinpointed predictors considering kidney function and its pregnancy-related physiologic adaptation for the prediction of PE.

We analyzed the predictive usefulness of GA-specific SCr distribution by comparing four combinations of feature sets. At first, SCr alone was included as an independent variable with covariates, such as age, labor type, HTN, DM, week of gestation at birth, and annotated as “SCr”. Secondly, SCr and its measured time (Mt) based on GA were implemented to establish the prediction model for adverse pregnancy outcomes, defined as “SCr + Mt”. We categorized the SCr levels into four categories (GAQ) based on the 25, 50, and 75 percentiles of SCr in each GW (GA-specific SCr distribution) [16,18], which were used to build the prediction model with SCr, defined as “SCr + GAQ”. Finally, all variables were included in the prediction model and defined as “SCr + Mt + GAQ”.

### 2.3. Establishment of the PE Prediction Model and Statistical Analysis

To validate the four combinations of feature sets, we conducted experimental tasks considering a random sampling perspective [23,24,25]. In studies identifying molecular signatures or clinical predictors related to disease or phenotype, comparative analyses among several lists of candidate gene sets or predictors were conducted based on random sampling. Two main tasks for the random validation strategy were performed: one was a random sample set [26] (e.g., iteration of random division of all data into training and testing sets) and the other was a random feature set [27,28]. Several studies used both random sample set and feature set to validate a disease-related signature [23,24]. For the random feature set, numerous variables were needed; however, our data only included finger-countable background features. Therefore, we only performed the random sampling to validate the four combinations of feature sets as follows:
(Step 1) We randomly divided the dataset using ratios of 0.7 and 0.3 and categorized them into training and testing sets, respectively.(Step 2) Using the candidate feature set (one of four feature sets), the training dataset, and logistic regression as the input variable, dataset, and classifier, respectively, we established the prediction model and measured the classification performance of adverse pregnancy outcomes in the testing dataset.(Step 3) For the same feature set, dataset, and classifier, we simultaneously measured the prediction performance of adverse pregnancy outcomes in the training dataset.(Step 4) We iterated steps 1 to 3 100 times, resulting in 100 pairs of predictive performance for the testing and training datasets.

For the comparative analysis of continuous variables (e.g., performance) according to different groups, a Student *t*-test or one-way analysis of variance were used. Two-tailed statistical tests were conducted, and a *p*-value less than 0.05 was considered statistically significant.

## 3. Results

### 3.1. General Characteristics of Participants Included in the PE Prediction Model

The mean age of pregnant women was 33.36 years (Table 1). There were 41.5% and 58.5% nullipara and multipara women, respectively. Before pregnancy, 1.5% of the patients were diagnosed with HTN while 4.0% had DM. Of the 1216 PE patients, 169, 114, 432, and 501 had PE alone without FGR and PTB; had both PE and FGR; had PE and PTB; and had PE, FGR, and PTB, respectively. The mean SCr level was 52.8 μmol/L (Table 1). Correlational analysis was applied to the anthropometric and laboratory markers, resulting SCr had direct or indirect relationships with blood urea nitrogen, liver profiles, and LDH (Appendix A).

Harel et al. [16] reported the GA-specific SCr distribution considering the pregnancy-related physiologic change, such as hyperfiltration. Our research team previously found that the GA-specific SCr distribution-based categorization of pregnant women could predict the adverse pregnancy outcomes [18]. Therefore, we described the demographic and medical characteristics according to the GA-specific SCr distribution (Appendix A). In line with previous study categorization of SCr measurements based on GA-specific SCr distribution [18], the following differential characteristics were shown as increment of GAQ: older age, higher ratios of multipara, HTN, DM, body mass index (BMI), and adverse pregnancy outcomes (Appendix A). These findings were consistent with those of our previous sample-based study [18]. 

### 3.2. Prediction Performance for PE

In the expert panel including domain experts (i.e., an obstetrician), a database administrator, a laboratorian, and a data scientist, we agreed that the measurement period of SCr of less than 38 weeks was clinically meaningful for predicting PE. Therefore, among 10,126 SCr measurements, cases measured at less than 38 GW were selected. As a result, among 1126 (PE any) and 4189 (Control) SCr measurement cases, 571 and 2625 were selected as PE and matched control groups, respectively. The 571 cases included participants with PE alone and all cases with PE and other adverse pregnancy outcomes, therefore annotated as “PE-any”. We compared performances of different feature sets (e.g., SCr, SCr + Mt, SCr + GAQ, and SCr + Mt + GAQ) for predicting the PE in any case using the random sampling method (Methods section). In classifying PE, the integration of Mt to SCr outperformed the models using only SCr levels. When considering the GA-specific SCr distribution, the prediction performance of PE was significantly improved compared to models using SCr only and SCr + Mt. The inclusion of all features (i.e., SCr + Mt + GAQ) provided the most accurate results for predicting PE (Figure 1).

Among 169 patients who exclusively had PE (PE-only) among the three adverse pregnancy outcomes (PE, PTB, and FGR), participants (n = 84) having SCr examined at less than 38 GW were selected to establish PE prediction model. As the matched control group (n = 2625), the SCr measurement cases from pregnant women without a pregnancy outcome were selected. The addition of GAQ did not improve the predictive performance for classifying PE, compared to models using only SCr levels. The inclusion of Mt did not also improve the accuracy of the model for predicting PE-only cases (Figure 2).

Next, we categorized cases diagnosed as PE and early PTB into the disease group. In this simulation, the multidisciplinary expert panel concluded that cases, including SCr measured at less than 34 GWs, were clinically useful for predicting early PTB. Therefore, 86 (PE + early PTB) and 1263 cases were categorized as disease and matched control groups, respectively. When predicting PE and early PTB, Mt was an important factor; moreover, GAQ did not improve the predictive performance of the 86 PE plus early PTB cases (Figure 3).

Participants with PE and late PTB were grouped into the disease group. We determined that SCr examinations at less than 37 GWs were eligible for this experiment, yielding 126 and 2218 cases as disease and matched control groups, respectively. The prediction of pregnancies with PE and late PTB exhibited an improved performance using Mt information (Appendix A), similar to that predicting PE and early PTB. When classifying cases with PE and FGR, Mt information provided better performance than GAQ, similar to experiments predicting PE plus early or late PTB (Appendix A).

Finally, we grouped pregnancies with PE and other adverse pregnancy outcomes (i.e., PTB and FGR) into the disease group (n = 248). The matched control group (n = 2625) was the same as that predicting the first simulation (Figure 1). Therefore, the Mt did not improve the predictive performance for the triple adverse pregnancy outcomes. Moreover, the GAQ ameliorated the classification performance for pregnancies with all types of adverse pregnancy outcomes (Figure 4).

### 3.3. Predictive Performance for PE According to GW

Varied simulations predicting PE were conducted according to the types of comorbidity, including PTB and FGR. It was considered that the GAQ provided augmented performances for predicting any case of PE, as well as the triple adverse pregnancy outcomes (PE + PTB + FGR). Therefore, we conducted a comparative analysis of the predictive accuracy in the aforementioned two cases (any PE or triple cases of adverse pregnancy outcomes), according to the GW. As a result, SCr examinations in the second trimester were well predicted, compared to those in the first or third trimester (Figure 5). Specifically, SCr measurements from 16 to 19 GWs suggest the best predictive performance for PE.

### 3.4. Final PE Prediction Model

We selected PE_any_ as the dependent variable and set SCr, Mt, GAQ, labor type, and chronic diseases as covariates in the final PE model. The parameters for each predictor were determined using a logistic regression analysis. To select the generalized weight values, we randomly sampled data from the entire dataset, corresponding to 70% of the data at 1000 iterations, thus creating 1000 training datasets. Using these training sets, 1000 lists of beta coefficients were generated, signifying that each predictor exhibited a Gaussian distribution with 1000 parameter values (following the central limit theorem). The mean value from this distribution was set as the final parameter.

The output values of the logistic regression analysis were obtained using a linear unit and a sigmoid function, which produced values between 0 and 1. To determine a generalized cutoff value for the output from a logistic regression-based network, we randomly sampled 70% of the entire dataset 1000 times to create 1000 test sets. By applying the final model to these 1000 datasets, we obtained the F-score distribution and 1000 values corresponding to the maximum F-score. Subsequently, by averaging these values, we obtained an optimal cutoff value of 0.262.

We applied the final PE model and optimal cut-off obtained from the logistic regression model (Figure 6) to the entire dataset, then analyzed the performance of classification model by four indices (Table 2). The current PE model exhibited exceptional performance in sensitivity and positive predictive value (PPV); however, it fell short in delivering satisfactory results in specificity and negative PV (NPV). In the case of applying the PE model to cases measured between 16.1 and 19 GWs, the sensitivity and PPV slightly decreased; moreover, the specificity and NPV notably increased compared to those obtained from all cases (Table 2).

## 4. Discussion

The current study that predicted PE demonstrated that the incorporation of knowledge domain (i.e., pregnancy-related renal hyperfiltration) improved the predictive accuracy. Specifically, when predicting PE, any case and triple cases (PE + PTB + FGR), while adding GA-specific SCr information, provided an enhanced predictive performance. Recent studies demonstrated that the dysregulation of pregnancy-related physiologic adaptation could predict adverse pregnancy outcomes based on statistical validation [17,18]. Our study identified that the GA-specific SCr distribution has predictive utility for PE based on a machine-learning-based method.

When predicting PE + PTB, the addition of GA-specific Cr information did not significantly improve the predictive power for PE and early (PE + PTB_early_) or late PTB (PE + PTB_late_). Moreover, the addition of Mt information still resulted in an improved predictive accuracy for PE + PTB. PTB has multiple factors contributing to its pathogenesis with progression beyond kidney-related biologic pathways, and our data may not have sufficient information on all these risk factors [29].

In our research findings, when including physiologic alteration in pregnant women, the significant improvement in predictive power was observed for the triple cases, which clinically refer to PTB and FGR, accompanied by PE [30]. A triple case of PE with concomitant PTB occurring before 37 weeks and FGR could be considered as an early-onset phenotype, as well as a severe form of PE. Therefore, it is important that the predictive power for the early and severe form of PE (triple case) be increased when incorporating SCr levels and GA-specific SCr distribution.

The current study showed that the SCr value obtained from blood sampling during the 2nd trimester of pregnancy, especially at 14–16 weeks of gestation, exhibited the best predictive power, which has significant clinical implications. Pregnant women are recommended to regularly visit the hospital for the antenatal check-up once every 4 weeks until the GA of 28 weeks [31,32,33]. At every 4-weekly visit, a medical history review, such as a review of system and physical examination, and ultrasound examination are usually conducted; however, blood tests are not typically performed in these routine check-ups. A blood test, which is included in the “integrated test” for screening fetal aneuploidy and neural tube defects, is typically conducted in two time periods: 11 to 13 and 14 to 22 GWs [33]. Moreover, a GA of 15 weeks is the best timing for secondary test to screen these diseases. When examining the integrated test, clinical, imaging, and laboratory tests are performed. The current study reported the SCr levels measured between 14 and 16 GWs to provide the most accurate predictive values (Figure 5). This period overlapped the secondary integrated test period. Our study is noteworthy in that the PE prediction model can be applied without substantially modifying the antenatal care and routine examination framework. Furthermore, the administration of aspirin to high-risk pregnant women at 16 GWs or less has a preventive effect against PE and its progression to severe PE [7,34,35]. Taken together, with the ease of the clinical application of our prediction model in real-world clinical setting and the potential for additional drug therapy, this study could contribute importantly to the preservation of the lives of pregnant women and fetuses.

In PE, glomerular endotheliosis gives rise to renal dysfunction due to the disruption of the endothelium and injury to the podocytes [11,36]. Renal damage manifests as HTN, decreased GFR, proteinuria, and SCr elevation. The kidney is one of the organs most affected by both normal pregnancy and PE. Normal accommodation includes a decreased systemic vascular resistance and glomerular hyperfiltration in healthy pregnant women. Thus, during pregnancy, as GA increases, GFR increases and SCr decreases. Recent molecular studies identified *CEBPB* and *GTF2B* as the core PE-related genes, which might be involved in extravillous trophoblast dysfunction in PE. Moreover, *GTF2B* was proposed as a potential cohub gene in Alzheimer’s disease and DM [37]. Therefore, it could be cautiously argued that PE is not just a uterine- and pregnancy-localized health issue, but a systemic disorder related to insulin resistance, mitochondrial dysfunction, and chronic inflammation.

Recently, placental growth factor (*PlGF*), soluble fms-like tyrosine kinase 1 (*sFlt-1*), and placental protein 13 (*PP13*) have been considered crucial biomarkers for the prediction and diagnosis of PE. *PIGF*, a member of the vascular endothelial growth factor (VEGF) family, is encoded in the human *PlGF* gene located on chromosome 14q14. *PIGF*, predominantly expressed in the placenta, is involved in angiogenesis, which is a crucial process for the growth and development of the fetus and maintenance of pregnancy [38]. *sFlt-1* is produced by the splicing of *FLT1* gene variant. Excess *sFlt-1* is known to disturb the action of VEGF, including angiogenesis, recruitment of endothelial progenitor cells, and endothelial integrity [39]. *PP13* is encoded in the placental gene, *LGALS13*, located in chromosome 19 at loci q13.2, and is mainly expressed in the testicular and brain tissues [40]. *PP13* is involved in placentation, immunoregulation, and regulation of blood pressure [41]. Individual levels of *PlGF*, *sFlt-1*, and *PP13* are reliable markers for the prediction and diagnosis of PE; however, their quantifications require immune-based techniques, such as enzyme-linked immunosorbent assays. In clinical settings, the aforementioned in-depth markers are difficult to measure, indicating the challenges to compare our PE model with those using deep phenotypical features. Our study used easily obtainable clinical and blood biomarkers, including SCr, its measured time, and the GW; therefore, it could provide a practical model that can be easily applied in primary clinical settings.

Kidney function or kidney disease is influenced by numerous factors [42]. In case of pregnancy, kidney function and its associated features form an intricate network with additional pregnancy-related factors [43,44]. Furthermore, the evaluation of kidney function using SCr is influenced by factors such as age and muscle mass, the measurement of which requires diagnostic tools, such as dual-energy X-ray and computed tomography scans. As cystatin C is less susceptible to other influences, it has been proposed as an alternative biomarker for assessing renal function in pregnant women [45]. However, in the clinical setting of obstetrics, including in university hospitals, collecting comprehensive and in-depth information on blood biomarkers or radiologic features, referred to as deep phenotyping [46], is a global challenge. This study aims to build a model based on easily collectible information in primary care settings. Other blood markers also undergo the adaptations related to pregnancy status [47]; therefore, adding extra blood markers to the PE prediction model requires prior studies analyzing the GW-specific distributions of each marker. Note that this study did not include several factors that affect kidney function; therefore, our PE model only plays a role as a watchman over whether PE may occur, and is meaningful as a supportive role for other validated diagnostic tools or markers.

The predictive performance for PE status decreased when using blood test results in first trimester (Figure 5). Several reasons, such as the relatively low number of SCr cases in the early stage of pregnancy and the early or transition stage of hyperfiltration, could explain the low predictive power. The data collected from the blood tests of pregnant women analyzed in the current study predominantly focused on the third trimester (Table 1). Tertiary hospitals, such as WSCH, primarily cater to women in the mid- to later stages of pregnancy who have been diagnosed with PE and referred from primary healthcare settings. Therefore, to collect information on pregnant women in their first trimester, it is crucial to implement the PE prediction model in primary healthcare settings. To facilitate the application of the model in primary healthcare settings, we developed a computationally efficient and highly interpretable model referred to as the shallow model (i.e., the logistic regression model) [48]. Furthermore, we created a classification model that incorporated the predictors readily available in primary care institutions (Figure 6). These efforts may contribute to the establishment of a generalized PE prediction model.

Our study had several limitations. Due to the retrospective design, the current dataset for the establishment of PE prediction model could not consist of crucial PE-related factors, such as medical treatment, blood pressure, autoimmune disease status, and biomarkers for diagnosing autoimmune disease, inflammatory markers, hemoglobin, and red and white blood cell count. Second, a modest fraction of anthropometric and laboratory markers, including liver profiles and body mass index (BMI), did not have complete data for all subjects (Appendix A), which meant their implementation into the PE prediction model was challenging. Furthermore, we could not fully dissect the correlational patterns among candidate biomarkers (e.g., obesity indices and anthropometric and laboratory markers) and the distributional changes based on gestational weeks (Appendix A). Therefore, in further studies, it will be required to establish the PE prediction model including comprehensive and in-depth biomarkers.

## 5. Conclusions

We demonstrated a significant improvement in the prediction of PE when SCr levels derived from a real-world hospital database and prior knowledge (renal physiological adaptation during pregnancy) were integrated into the predictive model. The implementation of both SCr and GAQ as predictors resulted in a substantial improvement in predicting PE_any_ and PE_triple_. We established a highly explainable practical model (i.e., logistic regression model) that can be easily applied in primary care settings without the need to modify the hospital database. Moreover, by determining the optimal cut-off, the model performance was comprehensively evaluated based on several performance indices, and high-risk PE groups could be screened.

## Figures and Tables

**Figure 1 biology-12-00816-f001:**
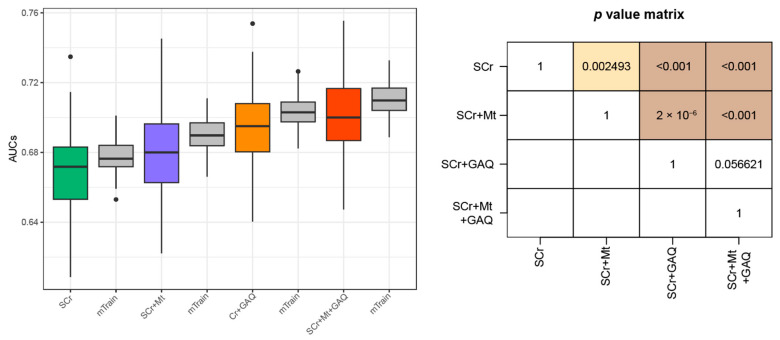
Prediction performances of all PE cases using the four feature sets. Any cases of PE (PE only + PE with other adverse pregnancy outcomes) were determined as the disease group. Prediction model was established using logistic regression. Boxplots include minimum, first quartile (25%), median, third quartile, and maximum values. Points in the boxplot referred to as outliers indicate cases showing more than 1.5 times the IQR, biased from the matched median value. Y-axis indicates AUC for predicting PE-any, and the box plot summarizes 100 levels of AUCs. Green-, purple-, orange-, and red-colored boxplots were obtained from the testing dataset, while grey-colored boxplots were curated from the training dataset. Abbreviations: PE, pre-eclampsia; SCr, serum creatinine; Mt, measured time; GAQ, quartile groups of gestational-age-specific SCr; IQR, interquartile range; AUC, area under receiver operating characteristic curve.

**Figure 2 biology-12-00816-f002:**
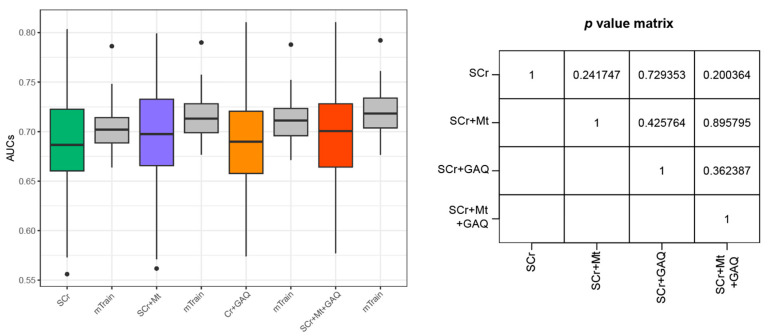
Prediction performances of PE-only case using the four feature sets. An exclusively diagnosed PE type among the three adverse pregnancy outcomes was defined as the disease group. Prediction model was established using logistic regression. Boxplots include minimum, first quartile (25%), median, third quartile, and maximum values. Points in the boxplot referred to as outliers indicate cases showing more than 1.5 times the IQR, biased from the matched median value. Y-axis indicates AUC for predicting PE-only, and the box plot summarizes 100 levels of AUCs. Green-, purple-, orange-, and red-colored boxplots were obtained from the testing dataset, while grey-colored boxplots were curated from the training dataset. Abbreviations: PE, pre-eclampsia; SCr, serum creatinine; Mt, measured time; GAQ, quartile of gestational-age-specific SCr; IQR, interquartile range; AUC, area under receiver operating characteristic curve.

**Figure 3 biology-12-00816-f003:**
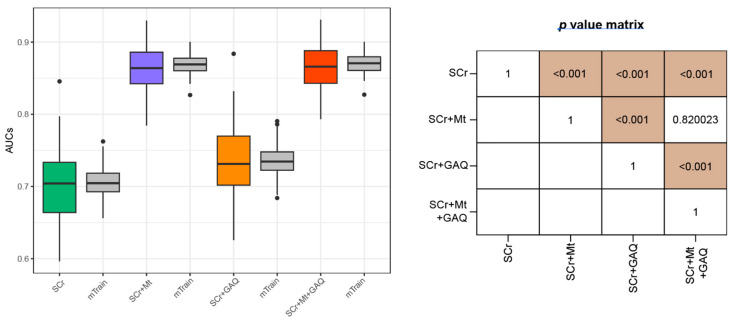
Prediction performances of PE with early PTB using the four feature sets. Pregnant women with PE and early PTB were categorized as the disease group. Prediction model was established using logistic regression. Boxplots include minimum, first quartile (25%), median, third quartile, and maximum values. Points in the boxplot referred to as outliers indicate cases showing more than 1.5 times the IQR, biased from the matched median value. Y-axis indicates AUC for predicting PE + PTB_early_, and the box plot summarizes 100 levels of AUCs. Green-, purple-, orange-, and red-colored boxplots were obtained from the testing dataset, while grey-colored boxplots were curated from the training dataset. Abbreviations: PE, pre-eclampsia; PTB, preterm birth; SCr, serum creatinine; Mt, measured time; GAQ, gestational-age-specific SCr quartile; IQR, interquartile range; AUC, area under receiver operating characteristic curve.

**Figure 4 biology-12-00816-f004:**
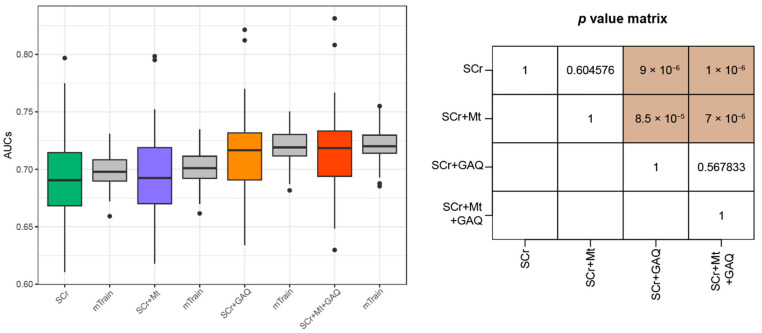
Prediction performances of the triple (PE + PTB + FGR) outcomes using four feature sets. Cases with the triple adverse pregnancy outcomes of PE, PTB, and FGR was determined as disease group. Prediction model was established using logistic regression. Boxplots include minimum, first quartile (25%), median, third quartile, and maximum values. Points in the boxplot referred to as outliers indicate cases showing more than 1.5 times the IQR, biased from the matched median value. Y-axis indicates AUC for predicting PE + PTB + FGR, and the box plot summarizes 100 levels of AUCs. Green-, purple-, orange-, and red-colored boxplots were obtained from the testing dataset, while grey-colored boxplots were curated from the training dataset. Abbreviations: PE, pre-eclampsia; PTB, preterm birth; FGR, fetal growth restriction; SCr, serum creatinine; Mt, measured time; GAQ, quartiles of gestational-age-specific SCr; IQR, interquartile range; AUC, area under receiver operating characteristic curve.

**Figure 5 biology-12-00816-f005:**
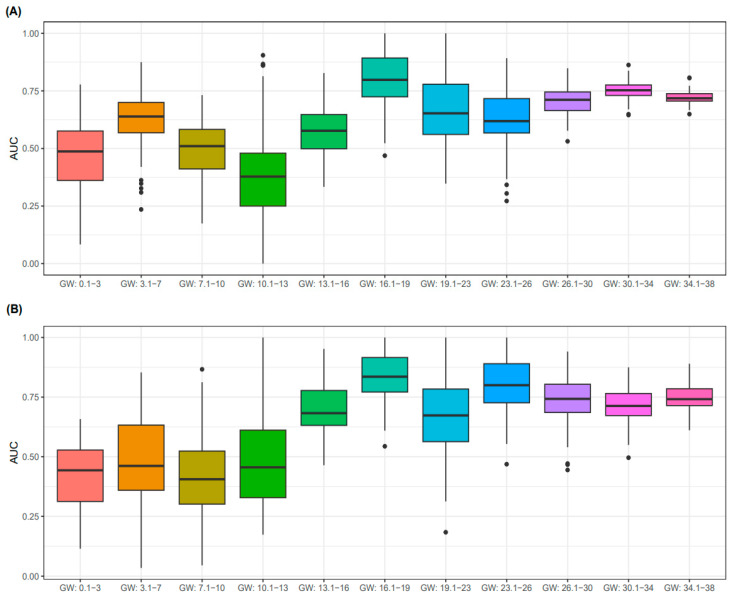
Prediction performances of PE according to GW. The prediction performances of PE-any (**A**) and PE-triple (**B**) cases are summarized according to GWs. Prediction model was established using logistic regression. Features for the model are SCr, Mt, and GAQ. Boxplots include minimum, first quartile (25%), median, third quartile, and maximum values. Points in the boxplot referred to as outliers indicate cases showing more than 1.5 times the IQR, biased from the matched median value. Y-axis indicates AUC for predicting PE, and the box plot summarizes 100 levels of AUCs in each GW category. Green-, purple-, orange-, and red-colored boxplots were obtained from the testing dataset, while grey-colored boxplots were curated from the training dataset. The PE-triple case denotes patients with PE, PTB, and FGR. Abbreviations: PE, pre-eclampsia; PTB, preterm birth; FGR, fetal growth restriction; GW, gestational week; AUC, area under receiver operating characteristic curve.

**Figure 6 biology-12-00816-f006:**
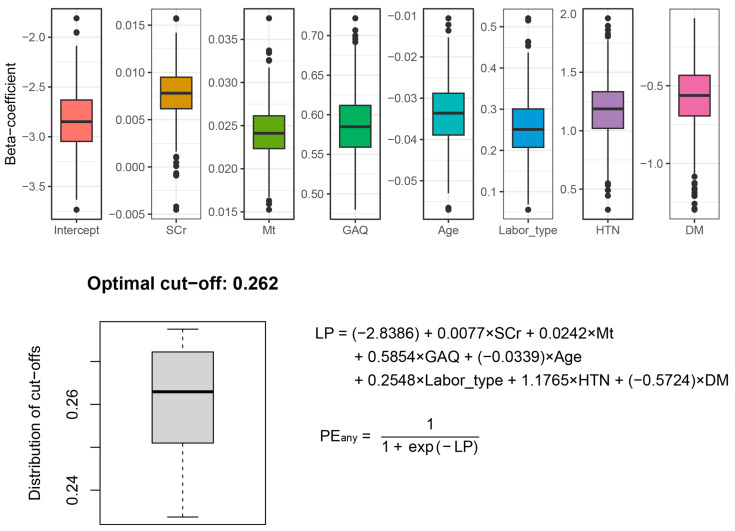
Final PE prediction model. Abbreviations: PE, pre-eclampsia; PTB, preterm birth; SCr, serum creatinine; Mt, measured time; GAQ, quartiles of gestational-age-specific SCr; HTN, hypertension; DM, diabetes; LP, linear predictor.

**Table 1 biology-12-00816-t001:** General characteristics of participants according to trimester.

	All	First Trimester GW < 14	Second Trimester 14 ≤ GW < 28	Third Trimester GW ≥ 28	*p*-for Trend
Number of SCr measurements	10,126	1302	1255	7569	
GW of SCr sampling, weeks (mean ± SE)	32.2 ± 0.12	7.3 ± 0.11	21.4 ± 0.12	38.2 ± 0.07	<0.001
GW of SCr sampling, weeks (median/IQR)	35.6 (27.7–40.1)	7.3 (4.1–10.6)	21.4 (17.7–25.3)	37.3 (34.7–40.3)	
Age, years (mean ± SE)	33.36 ± 0.05	33.6 ± 0.12	33.3 ± 0.13	33.3 ± 0.06	0.081
Labor types, n					
Nullipara	4205 (41.5)	430 (33)	506 (40.3)	3269 (43.2)	<0.001
Multipara	5921 (58.5)	872 (67)	749 (59.7)	4300 (56.8)	<0.001
Essential hypertension, n	153 (1.5)	12 (0.9)	28 (2.2)	113 (1.5)	0.024
Diabetes, n	407 (4.0)	70 (5.4)	71 (5.7)	266 (3.5)	<0.001
PE, n	1216 (12.0)	53 (4.1)	94 (7.5)	1069 (14.1)	<0.001
PE alone, n	169 (1.7)	17 (1.3)	9 (0.7)	143 (1.9)	0.006
PE + FGR, n	114 (1.1)	3 (0.2)	3 (0.2)	108 (1.4)	<0.001 ^a^
PE + PTB, n	432 (4.3)	15 (1.2)	27 (2.2)	390 (5.2)	<0.001
PE + FGR + PTB, n	501 (4.9)	18 (1.4)	55 (4.4)	428 (5.7)	<0.001
BMI, kg/m^2^ (mean ± SE) ^a^	26.6 ± 0.13	23.2 ± 0.32	24.1 ± 0.38	27.1 ± 0.14	<0.001
SCr, μmol/L (mean ± SE)	52.8 ± 0.38	55.2 ± 0.77	48.9 ± 1.22	53 ± 0.45	0.605
BUN, mg/dL (mean ± SE) ^b^	9.2 ± 0.05	9.8 ± 0.15	8.3 ± 0.17	9.2 ± 0.06	0.333
AST, U/L (mean ± SE) ^b^	28.1 ± 0.51	26.5 ± 1.99	26.3 ± 1.29	28.8 ± 0.55	0.06
ALT, U/L (mean ± SE) ^b^	22.4 ± 0.57	26.8 ± 2.94	21.5 ± 1.11	21.8 ± 0.52	0.009
ALP, U/L (mean ± SE) ^b^	126.1 ± 0.87	62.8 ± 0.97	73.9 ± 1.17	142.8 ± 1	<0.001
GGT, U/L (mean ± SE) ^b^	19.2 ± 0.41	24.9 ± 1.66	15.6 ± 0.59	18.9 ± 0.47	<0.001
LDH, U/L (mean ± SE) ^b^	229.2 ± 3.36	180.2 ± 2.28	206.2 ± 5.81	265.6 ± 5.69	<0.001

Continuous and categorical variables are presented as mean ± SE and number (percent), respectively. To analyze the linear trend in a continuous feature according to three GW groups, 1, 2, and 3 values were arranged as the representative values for the first, second, and third trimester groups, respectively, then implemented as independent variables in one-way analysis of variance. ^a^ *p*-value for PE + FGR was calculated using Fisher’ exact test, and others were measured using *t*-test or chi squared test. ^b^ The summary statistics were measured based on cases less than number of all subjects (n = 10,126) due to some missing values. Abbreviations: GW, gestational week; SCr, serum creatinine; SE, standard error; IQR, interquartile range; PE, pre-eclampsia; FGR, fetal growth restriction; PTB, preterm birth; BMI, body mass index; BUN, blood urea nitrogen; AST, aspartate aminotransferase; ALT, alanine aminotransferase; ALP, alkaline phosphatase; GGT, gamma-glutamyl transferase; LDH, lactate dehydrogenase.

**Table 2 biology-12-00816-t002:** Performance of pre-eclampsia prediction model.

Group	Sensitivity	Specificity	PPV	NPV
All	0.79	0.51	0.881	0.346
GW (16.1–19)	0.745	0.619	0.837	0.481

The 0.262 (optimal cut-off) obtained from logistic regression model (Figure 6) was implemented to evaluate the performance of PE prediction model based on four indices (sensitivity, specificity, PPV, and NPV). Abbreviations: GW, gestational week; PPV, positive predictive value; NPV, negative predictive value.

## Data Availability

Detailed data that support the findings of this research are not available because of privacy concerns and hospital regulation restrictions to protect patients. Anonymized data may be available with the permission of the corresponding author upon reasonable request.

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
