# Peer review of "Prediction Model for Pre-Eclampsia Using Gestational-Age-Specific Serum Creatinine Distribution"

_biology, 2023, doi:10.3390/biology12060816_

Round 1
Reviewer 1 Report
This manuscript sounds good and contains significant scientific data. There are certain points that need careful attention to address the following points:
1. The data presented in the simple summary should be different from that presented in the abstract section. Authors have repeated certain sentences in both sections.
2. In the introduction section, authors should develop a correlation between pre-eclampsia with serum creatinine during the pregnancy by presenting the experimental and epidemiological studies.
3. Did the authors record the term of trimester during the recruitment of the participants.
4. In the methodology section, there is no need to add the “Definitions of adverse pregnancy outcomes”.
5. The conclusion section should be re-written, and the key findings should be summarized.
6. There are certain grammatical mistakes and syntax errors.
6. There are certain grammatical mistakes and syntax errors.
Author Response
We respond the Reviewer's comment using PDF file.

Reviewer 2 Report
To authors,
Many predictive methods have been proposed to predict the occurrence of PE. The present method is less expensive and provided good predictive value. Although whether this model actually is of significant use in a real-world practice, and if yes, whether it holds true to any other population (racial differences) are to be clarified, the present data is a gigantic leap as a novel methodology to predict PE.
Author Response

(The authors gave the same response as above.)

Reviewer 3 Report
The manuscript describes a novel prediction method of preeclampsia utilizing gestational age-specific serum creatinine. The manuscript is well structured and was a pleasure to read. Overall, the topic is very relevant, and deserves to be eventually published. Specific comments:
Methods:
1. The authors reported on 2 variables namely the coexistence of HTN and DM. But renal function could be altered due to a variety of confounders such as renal pathology, autoimmune conditions, medication use, BMI etc. Are the authors capable on reporting also on other important confounders.
2. The authors do not specifically state when the sCr samples were taken. Please elaborate on when sampling was done.
3. If the authors want to speculate on the prognostic value of sCr then the authors should propose a specific cutoff (based on AUC curve) with given sensitivity, specificity, negative and positive predictive value. And if possible, compare this to the current gold standard for prediction of preeclampsia.
Results:
4. In table 1 the authors could also give the mean GA of sCR sampling. Ideally an overview per GA when the samples were taken and the mean results thereof.
5. If the authors want to speculate on the predictive value of sCR the results of only AUC would not suffice. Please report on sensitivity, specificity, positive and negative predictive values specifically for the subgroup of samples taken at 16-19w.
Discussion
6. Ultimately it is not clear to the reader whether the authors actually described a prognostic rather than predictive biomarker. Since it appears that the vast majority of samples were taken after 34w. Please elaborate in the discussion.
7. In the discussion the authors should highlight the problem of not accounting for other confounders and variations of renal function.
8. Line 327. Please delete the duplicated sentence.
9. For this marker to be incorporated into clinical setting first the true value of a specific cutoff has to be proposed and next it needs to be prospectively validated.
Author Response

(The authors gave the same response as above.)

Round 2
Reviewer 3 Report
Thank you, for the amendments, The authors have adressen all my major concerns.
Author Response
Answers to comments by editor (for comments presented by reviewer #3)
1.1. The authors reported on 2 variables namely the coexistence of HTN and DM. But renal function
could be altered due to a variety of confounders such as renal pathology, autoimmune conditions,
medication use, BMI etc. Are the authors capable on reporting also on other important
confounders.
1.2. Question 1: it is absolutely possible to find the clinical serum data on other markers in these
patients, as it is a normal routine test (editor).
Answer) Additional discussions are always welcome. Your opinion is an accurate comment that can
only be made by expert who truly understands the practical aspects of the medical field. Strictly
speaking, an accurate method for measuring renal function involves administering inulin or
radiolabeled isotopes and observing the amount and rate of their excretion in real-time. However, the
methods for measuring renal function requires a significant amount of time and expense, and can be
particularly challenging to perform these tests in pregnant women. That is why estimating renal
function through tests such as serum creatinine (SCr) or cystatin C is commonly used. However, SCr is
significantly influenced by muscle mass, which requires diagnostic tools, such as Dual-Energy X-ray
Absorptiometry (DEXA) for its measurement. Furthermore, there is a significant lack of widely
accepted equations to derive a SCr adjusted by muscle mass. I have not added specific details before,
and in order to convey the meaning more accurately, I have further supplemented the above contents
in discussion section.
Secondly, adding other markers to the current data without a specific strategy should be
approached with caution. Indeed, other markers also undergo adaptation-related to pregnancy status
[1], and most of these markers do not have a sufficient amount of big data generated to establish the
pregnancy-related adaptation distributions. Therefore, adding other markers without a rigorous
strategy can potentially deteriorate the performance of predictive models. Even if there is an
improvement in performance, the interpretability of the model may significantly decrease. It is deemed
necessary to establish an adequate reference range for candidate markers based on gestational age
before incorporating additional markers into the predictive model. We also added this information to
the discussion section.
7
th paragraph of Discussion section)
Kidney function or kidney disease is influenced by numerous factors [42]. In case of pregnancy,
kidney function and its associated features form an intricate network with additional pregnancyrelated factors [43,44]. Furthermore, evaluation of kidney function using SCr is influenced by factors
such as age and muscle mass of which measurement requires diagnostic tools, such dual-energy Xray and computed tomography scan. As cystatin C is less susceptible to other influences, it has been
proposed as an alternative biomarker for assessing renal function in pregnant women [45]. However,
in the clinical setting of obstetrics, including in university hospitals, collecting comprehensive and
in-depth information on blood biomarkers or radiologic feature, referred to deep phenotyping [46],
is a global challenge. This study aims to build a model based on easily collectible information in
primary care settings. Other blood markers also undergo the adaptations-related to pregnancy status
[47], therefor, adding extra blood markers to the PE prediction model requires prior studies
analyzing GW-specific distributions of each marker. Note that this study did not include several
factors that affect kidney function, therefore, our PE model only plays a role as a watchman that PE
may occur, and is meaningful as a supportive role for other validated diagnostic tools or markers.
2.1. The authors do not specifically state when the sCr samples were taken. Please elaborate on when
sampling was done.
2.2. Question 2: The authors did not correctly answer the question. The question is when the samples
were collected.
Answers) Your comment has made us aware that I omitted an important point. To be precise, SCr
(serum creatinine) was measured during early, mid, and late pregnancy prior to childbirth. The blood
samples were collected from pregnant women during their prenatal check-ups at the hospital, then,
these samples were used to predict the occurrence of preeclampsia (PE) in the future. The timing of
initial visits for pregnant women can vary individually, and blood samples may be collected during the
first trimester of pregnancy. In cases that are referred from a primary healthcare provider or local clinic,
the measurements could have been taken during the second trimester. As part of routine antenatal care,
following the commonly practiced guidelines, all pregnant women undergo screening tests, including
various hematological analyses, including SCr, during the third trimester of pregnancy before
childbirth. We supplemented the above contents in method section.
2.1 section in manuscript)
Blood test, such as SCr was conducted during early, mid, and late periods of pregnancy. The timing
of initial visits for pregnant women varies individually. In cases that are referred from a primary
local clinic, the first blood test is conducted during the second or third trimester. Most pregnant
women undergo systematic blood test during the third trimester of pregnancy before childbirth.
3.1. If the authors want to speculate on the prognostic value of sCr then the authors should propose
a specific cutoff (based on AUC curve) with given sensitivity, specificity, negative and positive
predictive value. And if possible, compare this to the current gold standard for prediction of
preeclampsia.
3.2. Question 3: the authors also did not correctly or well answer the question.
Answer) The comment consists of two aspects that have been integrated. The first aspect suggests using
a cutoff point and evaluating the performance through various metrics, and we responded to this by
providing additional figures and tables.
The second aspect, which the editor may be pointing out, is about the recent introduction of
prominent PE prediction models. However, most prominent models for PE status require in-depth
markers, such as serum PLGF and serum sFLT-1 to calculate the model's output. Therefore, it is almost
impossible to directly compare our study with existing models. Additionally, to the best of our
knowledge, there is no research that calculates actual PE prediction values between 0 and 1 using SCr
values and hyperfiltration distribution. We have included this information as an additional point in our
discussion.
6
th paragraph in discussion section)
In clinical settings, the aforementioned in-depth markers are difficult to be measured, indicating the
challenges to compare our PE model with those using the deep phenotypical features.
4.1. In table 1 the authors could also give the mean GA of sCR sampling. Ideally an overview per
GA when the samples were taken and the mean results thereof.
4.2. Question 4: Please change gestational week to gestational week at sampling (if I am correct) in
Table 1.
Answer) I apologize for the mistake. We have revised the mentioned part.
5. If the authors want to speculate on the predictive value of sCR the results of only AUC would
not suffice. Please report on sensitivity, specificity, positive and negative predictive values
specifically for the subgroup of samples taken at 16-19w.
Answer) OK
6.1. Ultimately it is not clear to the reader whether the authors actually described a prognostic rather
than predictive biomarker. Since it appears that the vast majority of samples were taken after
34w. Please elaborate in the discussion.
6.2. Question 6: I strongly disagree with the answer, even though I am happy with the sample size
in the first and second trimesters.
Answer) I apologize for the difficulty in providing an exact answer. Firstly, our study resulted that the
predictive power decreases when using SCr values in early pregnancy. There can be several reasons for
the decreased accuracy in PE prediction using early pregnancy blood tests. One reason may be the
limited availability of absolute numbers in early pregnancy blood test results. Secondly, since
hyperfiltration is not prominent during early pregnancy, it may result in a lack of predictive value. Your
comment is also valid, as the best performance was provided during the 16th to 19th week of pregnancy,
a period when the numerical values in blood test results are not significantly abundant. We are grateful
for your insight, as it has allowed us to observe and interpret our results in more detail. We included
the aforementioned points in discussion section.
8
th paragraph in discussion section)
The predictive performance for PE status decreased when using blood test results in first trimester
(Figure 5). Several reasons, such as relatively low number of SCr cases in early stage of pregnancy
and the early or transition stage of hyperfiltration could explain the low predictive power.
7.1. In the discussion the authors should highlight the problem of not accounting for other
confounders and variations of renal function.
7.2. Question 7: see my comments on Question 1.
Answer) We respond this comment in the first part.
Reference
1. Teasdale, S.; Morton, A. Changes in biochemical tests in pregnancy and their clinical
significance. Obstet Med 2018, 11, 160-170, doi:10.1177/1753495x18766170.
